# Nutraceutical Properties of Thai Mulberry (*Morus alba* L.) and Their Effects on Metabolic and Cardiovascular Risk Factors in Individuals with Obesity: A Randomized, Single-Blind Crossover Trial

**DOI:** 10.3390/nu16244336

**Published:** 2024-12-16

**Authors:** Wason Parklak, Monchai Chottidao, Narongsuk Munkong, Surat Komindr, Sudjai Monkhai, Bandhita Wanikorn, Niromlee Makaje, Kanokwan Kulprachakarn, Hataichanok Chuljerm, Surasawadee Somnuk

**Affiliations:** 1Research Center for Non-Infectious Diseases and Environmental Health, Research Institute for Health Sciences, Chiang Mai University, Chiang Mai 50200, Thailand; wason.p@cmu.ac.th (W.P.); kanokwan.kul@cmu.ac.th (K.K.); hataichanok.ch@cmu.ac.th (H.C.); 2College of Sports Science and Technology, Mahidol University, Nakhon Pathom 73170, Thailand; monchai.cho@mahidol.edu; 3Department of Pathology, School of Medicine, University of Phayao, Phayao 56000, Thailand; narongsuk.mu@up.ac.th; 4Division of Nutrition and Biochemical Medicine, Department of Medicine, Faculty of Medicine, Ramathibodi Hospital, Mahidol University, Bangkok 10400, Thailand; surat.kom@mahidol.ac.th; 5Wangnumkeaw Sub-District Health Promotion Hospital, Nakhon Pathom 73140, Thailand; wangnumkeaw@gmail.com; 6Department of Biotechnology, Faculty of Agro-Industry, Kasetsart University, Bangkok 10900, Thailand; agibtw@ku.ac.th; 7Department of Sports Science, Faculty of Sports and Health Science, Kasetsart University, Kamphaeng Saen Campus, Nakhon Pathom 73140, Thailand; niromlee.m@ku.th; 8School of Health Sciences Research, Research Institute for Health Sciences, Chiang Mai University, Chiang Mai 50200, Thailand

**Keywords:** Kamphaeng Saen mulberry, total phenolic compounds, total anthocyanins, obesity, metabolic risk factors, cardiovascular risk factors

## Abstract

**Background/Objectives**: Mulberries exhibit antioxidant properties that may attenuate metabolic abnormalities. Kamphaeng Saen mulberry (KPS-MB-42-1) contains anthocyanins, polyphenols, and nutrients, but few studies have explored its benefits for human health. This study investigated the effects of a concentrated mulberry drink (CMD) from the KPS-MB-42-1 cultivar on metabolic and cardiovascular risk factors in obese individuals. **Methods**: A single-blind, randomized crossover clinical pilot trial was performed on individuals with obesity. Participants consumed 100 g of CMD daily, alternating with placebo for 6 weeks. Body composition, blood pressure, and blood samples were assessed at baseline and post-intervention. **Results**: This study was completed with 12 participants (7 men, 5 women, aged 30–55 years, BMI 32.1 ± 5.98 kg/m^2^) consuming CMD with 1041.90 mg total phenolic compounds and 35.34 mg total anthocyanins. No significant changes in body composition were observed. CMD consumption significantly reduced systolic and diastolic blood pressure, and mean arterial pressure, compared to baseline and placebo periods (*p* < 0.05). While total cholesterol, LDL-C, and HDL-C remained unchanged, triglycerides were significantly lower during CMD consumption compared to placebo periods (*p* < 0.05). Fasting plasma glucose (FPG) levels were stable during CMD consumption but increased significantly with the placebo period (*p* < 0.05). C-reactive protein levels were also significantly lower during CMD consumption compared to placebo periods (*p* < 0.05). No changes in blood coagulation indicators (prothrombin time, activated partial thromboplastin time, and the international normalized ratio) were found. **Conclusions**: CMD improved metabolic markers, particularly regarding its antihypertensive effects. These findings highlight CMD’s potential as a health drink for managing metabolic syndrome and preventing chronic diseases.

## 1. Introduction

Obesity is a critical global health issue, and its prevalence continues to rise annually [1,2]. Central obesity, in particular, disrupts various body systems and is strongly associated with insulin resistance, increasing the risk of type 2 diabetes (T2D) and cardiovascular diseases (CVD) [3,4]. This condition also contributes to metabolic syndrome, characterized by a combination of central obesity, insulin resistance, elevated triglycerides, low HDL-C, and hypertension, often accompanied by chronic low-grade inflammation [5,6]. Obesity also induces systemic oxidative stress, driven by factors such as hyperleptinemia, tissue dysfunction, reduced antioxidant defenses, and chronic inflammation [6,7,8]. These factors promote atherosclerosis and abnormalities in blood coagulation, increasing levels of fibrinogen and factor VII while reducing anticoagulant proteins like protein C and antithrombin, thus elevating the risk of thrombosis [9,10].

Weight loss has been shown to reduce oxidative stress, enhance antioxidant protection, and mitigate pathological risk factors associated with obesity [11]. Unhealthy lifestyle behaviors are often linked to the primary causes of obesity. Therefore, the management of obesity primarily focuses on weight reduction through a combination of strategies, including reducing the intake of high-calorie foods, engaging in physical activity, and using pharmacological interventions to normalize fat metabolism [12]. Currently, research emphasizes the use of natural products rich in bioactive compounds due to their higher safety profiles compared to synthetic chemicals [13,14]. Phenolic compounds, a class of phytochemicals, have been shown to slow the progression of various diseases [15,16] and reduce the risk of non-communicable diseases [17]. Polyphenols specifically enhance insulin sensitivity [18], decrease lipid synthesis [19], and promote fatty acid oxidation (β-oxidation) [20,21]. Epidemiological evidence indicates that adequate polyphenol intake positively impacts blood pressure regulation, lipid profiles, and glycemic control [22].

Mulberry (*Morus alba* L.), widely cultivated in Asia, contains phytochemicals such as polyphenols (e.g., anthocyanins, flavonoids, and phenolic acids). Traditionally, mulberries have been used to manage diabetes [23], reduce cholesterol, combat obesity [24,25,26], and provide antioxidant effects [27]. Kamphaeng Saen mulberry (KPS-MB-42-1), a white mulberry cultivar, is particularly rich in polyphenols and anthocyanins [28,29,30]. Despite its potential in managing metabolic syndrome and chronic diseases, research on its health benefits remains limited. This study aims to investigate the effects of concentrated KPS-MB-42-1 mulberry drink on metabolic and cardiovascular risk factors in obese individuals, providing insights for further applications and research.

## 2. Materials and Methods

### 2.1. Preparation of Concentrated Mulberry Drink

This research employed the Kamphaeng Saen mulberry-42-1 (KPS-MB-42-1) variety, obtained from farms in the rural regions of Kamphaeng Saen District, Nakhon Pathom, Thailand. Preliminary laboratory investigations demonstrated that heat treatment of KPS-MB-42-1 mulberries markedly enhanced polyphenol content relative to untreated mulberries. This study formulated a concentrated mulberry drink based on these findings.

The procedure entailed decocting mulberries in water at a 2:1 fruit-to-water ratio for around 15 min. The flavor of the beverage was modified with maltitol as a sweetening agent. The produced beverage was thereafter enclosed in sterilized screw-cap bottles. Each container of concentrated mulberry drink was standardized to contain an equal portion of 50 g per bottle. The formulation and production processes were carried out at the Division of nutrition for Health and Sports, Department of Sports Sciences, Faculty of Sport and Health Science, Kasetsart University, Kamphaeng Saen Campus.

The nutritional composition and bioactive compounds of mulberry drink were examined at the Department of Biotechnology, Faculty of Agro-Industry, Kasetsart University, Bangkok, Thailand, employing the following methodologies: Proximate composition analyses were conducted in accordance with AOAC procedures. The sugar profiles in the drink were analyzed through High-Performance Liquid Chromatography (HPLC). The content of total phenolic compounds was assessed utilizing the Folin–Ciocalteu assay. The quantification of total anthocyanin content was conducted using colorimetric methods.

### 2.2. Study Subjects and Design

This clinical trial was a pilot study conducted at the College of Sports Science and Technology, Mahidol University, Thailand, from June to October 2023. This study adhered to the principles of the Declaration of Helsinki, and all procedures involving human participants were approved by the Mahidol University Multi-Faculty Cooperative IRB Review (COA no. MU-MOU 2023/104.1007).

After receiving approval from the Institutional Review Board for this study, the researchers utilized posters to announce and recruit subjects interested in participating in the project. All interested individuals were required to meet the inclusion criteria specified by this study’s conditions. The inclusion criteria for subjects in this study required individuals to be aged 30–55 years, with a body mass index (BMI) greater than 25 kg/m^2^, and central obesity (waist circumference exceeding 90 cm in males and 80 cm in females). Participants also needed to have a systolic blood pressure (SBP) greater than 130 mmHg and/or a diastolic blood pressure (DBP) greater than 85 mmHg. The exclusion criteria included individuals with liver disease, kidney disease, heart disease, thyroid disorders, adrenal gland disorders, or cancer. Subjects were excluded if they had a history of antibiotic use within three months prior to this study, consumed alcoholic beverages 2–3 times per week, or had severe metabolic syndrome-related conditions such as cardiovascular disease, diabetes, or non-alcoholic fatty liver disease (NAFLD). Additionally, individuals who consumed beverages or supplements containing polyphenols, those receiving immunosuppressive medications (e.g., steroids), or those with gastrointestinal disorders such as chronic constipation, diarrhea, inflammatory bowel disease, or other chronic gastrointestinal issues were excluded. Women who were pregnant or breastfeeding were also not eligible for this study.

Subjects received comprehensive information about the research project and an informed consent document seven days before this study started (baseline period). Baseline data collection during this period encompassed demographic information, including age, sex, height, weight, waist circumference, hip circumference, and blood pressure, as well as medical history. Subjects who fulfilled the inclusion criteria and consented to participate signed the informed consent form. Subsequently, body composition was assessed utilizing InBody 720 (Biospace Dba InBody, California, CA, USA). Blood samples were obtained to assess fasting blood glucose, insulin levels, the homeostatic model assessment of insulin resistance (HOMA-IR) index, lipid profile, inflammatory markers, and coagulation parameters (coagulogram). The HOMA-IR index was calculated using the formula [31]:HOMA-IR = [fasting insulin (µU/mL) × fasting glucose (mg/dL)]/405(1)

Subjects were required to undergo a 10 h fasting period prior to blood collection. Additionally, subjects completed a 24 h food recall over three non-consecutive days (two weekdays and one weekend day). The recorded dietary data were analyzed using the INMUCAL-Nutrients V.4.0 software (Mahidol University, Nakhon Pathom, Thailand) to calculate the average daily nutrient intake.

This study was conducted as a randomized single-blind crossover trial. A total of 22 subjects were screened after being fully informed of the trial procedures, and ultimately, 15 subjects were enrolled in this study. Subjects were randomly assigned to receive either a concentrated mulberry drink (CMD) or a placebo drink in a crossover manner. CMD was administered at a daily dose containing 35 mg of total anthocyanins, a level consistent with previous studies that demonstrated improvements in metabolic markers [32]. The daily dose was divided into two servings, taken after the main meals (breakfast and dinner), with a total volume of 100 g (2 bottles per day). The placebo drink contained no mulberry extract but was formulated to provide an equivalent caloric content to CMD. The intervention period lasted for six weeks, in line with similar studies [33,34,35]. At the end of each allocation period, participants underwent a comprehensive physical assessment and blood sample collection, with analyses conducted at Chandrubeksa Hospital Laboratory, Kamphaeng Saen District, Nakhon Pathom. Blood samples were analyzed for various parameters as in the baseline period. Following the first allocation period, a two-week washout period was implemented to minimize carryover effects. Subjects then crossed over to the alternate drink for the second allocation period. The randomized single-blind crossover design of this study is depicted in Figure 1.

### 2.3. Statistical Analysis

Continuous variables were presented as mean ± standard deviation (SD), while nutritional data and bioactive compound content of CMD were summarized using mean values alone. Differences across time points within each intervention group were evaluated using two-tailed paired Student’s *t*-tests. Statistical analyses were carried out using SPSS software, version 15.0 (SPSS Inc., Chicago, CA, USA).

## 3. Results

### 3.1. Concentrated Mulberry Drink

According to Table 1, 100 g of CMD has 78.08 kilocalories. It has 18.78 g of carbohydrates, mainly sugars, amounting to 18.74 g, which include fructose (2.51 g), glucose (3.07 g), and maltose (13.16 g). Furthermore, it contains 0.12 g of fat and 0.47 g of protein. The drink comprises 1041.9 mg of total phenolic compounds and 35.34 mg of total anthocyanins.

### 3.2. Baseline Characteristics of Subjects

A total of 12 subjects were selected according to the study criteria and completed this study. Of these, seven were male, accounting for 58.3% of the sample, and five were female, representing 41.7%. In Table 2, the baseline physical characteristics of this study’s subjects are presented. The average age of the subjects was 46.57 ± 8.49 years, ranging from 30 to 55 years. The subjects’ mean body weight was 92.1 ± 15.55 kg, and the average body mass index (BMI) was 32.1 ± 5.98 kg/m^2^, indicating a range that included values consistent with overweight or obesity. The mean muscle mass was 33.7 ± 6.45 kg, and the average fat mass was 32.2 ± 12.26 kg. The percentage of body fat was 34.42 ± 10.07%. Waist circumference (WC), a critical measure associated with abdominal obesity, averaged 110.83 ± 16.80 cm for males and 110.00 ± 30.33 cm for females. The waist-to-hip ratio (WHR) averaged 0.99 ± 0.03 for males and 0.98 ± 0.05 for females, both indicating central obesity trends. Additionally, the average visceral fat area level, which is highly correlated with obesity-related health risks, was 13.89 ± 5.06. The average systolic blood pressure (SBP) was 137.4 ± 12.28. The average diastolic blood pressure (DBP) was 92.56 ± 10.45 mmHg. According to this study’s criteria, elevated blood pressure is defined as SBP of ≥130 mmHg or DBP of ≥85 mmHg.

Table 3 presents the baseline blood chemistry characteristics of the subjects in this study. The triglyceride (TG) level of the participants was 215 ± 90.52 mg/dL. The total cholesterol (TC), low-density lipoprotein cholesterol (LDL-C), and high-density lipoprotein cholesterol (HDL-C) levels were 220.67 ± 44.87 mg/dL, 145.33 ± 36.35 mg/dL, and 42.89 ± 8.84 mg/dL, respectively. The average fasting plasma glucose (FPG) level was 108.89 ± 16.62 mg/dL, and the fasting plasma insulin (FPI) level was 11.38 ± 6.29 IU/mL, both within the normal range. However, the insulin resistance score, as measured by the homeostatic model assessment for insulin resistance (HOMA-IR), was 2.97 ± 1.39, indicating elevated insulin resistance.

The frequency of exercise or sports participation among the subjects indicated that the majority did not engage in sports or exercise, accounting for 75.0% of the group. Weekly exercise frequency was as follows: 8.3% exercised 1–2 days per week, 8.3% exercised 3–5 days per week, and 8.3% exercised or played sports every day, as shown in Figure 2.

### 3.3. Nutrition and Energy Intake During Intervention

The subjects in this study consumed 100 g of CMD daily, containing 1041.90 mg of total phenolic compounds and 35.34 mg of total anthocyanins (as presented in Table 1). The drink was administered in two servings per day, one in the morning and one in the evening, over a 6-week period. During the placebo phase, participants consumed a placebo beverage of identical volume (100 g per serving, twice daily) that contained no mulberry-derived compounds. This study employed a crossover design, allowing each participant to serve as their own control. The effects of consuming the CMD and the placebo were evaluated by comparison to the baseline period during which subjects consumed neither drink.

Table 4 and Table 5 present the nutrient and energy intake of twelve subjects, as recorded through a 24 h dietary recall. The food diary method, covering three days (two weekdays and one weekend day), was used to document the type and quantity of foods consumed. The subjects’ macronutrient intake, expressed as a percentage of total energy, showed that carbohydrates contributed 54.60–56.91%, fat contributed 27.14–29.51%, and protein contributed 15.37–16.11% of their daily energy intake.

The analysis revealed no significant differences in the amount of nutrients and energy consumed during the baseline period, the intervention with CMD, or the placebo phase. This consistency in dietary intake across all study periods ensures that the observed outcomes of the intervention were not influenced by variations in the subjects’ diet.

### 3.4. Effects of CMD Consumption on Body Composition

Table 6 shows that there were no statistically significant differences in body weight, BMI, muscle mass, WC, HC, or WHR when comparing the baseline period to the period after consuming the CMD or the placebo. Similarly, fat percentage (%), fat mass, and visceral fat area levels showed no significant changes across the baseline, CMD, and placebo phases. These results indicate that the consumption of the concentrated mulberry drink or placebo did not significantly affect body composition parameters during this study.

### 3.5. Effects of CMD Consumption on Changes in Blood Pressure, Heart Rate, and Mean Arterial Pressure in Subjects

After consuming CMD, subjects demonstrated a marked reduction SBP (as shown in Figure 3A). Specifically, SBP after CMD consumption was 127.77 ± 6.64 mmHg, which was significantly lower compared to the baseline value of 137.44 ± 12.28 mmHg (*p* < 0.05). Moreover, SBP during the CMD phase was also significantly lower than during the placebo phase, where the average SBP was recorded at 135.11 ± 7.68 mmHg (*p* < 0.05). Similarly, DBP was also reduced after CMD consumption (as shown in Figure 3B). The average DBP during the CMD phase was 86.77 ± 4.60 mmHg, which was significantly lower than the baseline value of 92.56 ± 10.45 mmHg (*p* < 0.05). Furthermore, DBP was significantly lower during the CMD phase compared to the placebo phase, where the value was 93.67 ± 4.97 mmHg (*p* < 0.05).

Regarding heart rate (HR), no significant changes were observed between the baseline, CMD, and placebo phases. As displayed in Figure 3C, the average HR during the CMD phase (76.00 ± 7.41 bpm) was similar to the baseline value (80.00 ± 11.12 bpm) and the placebo phase (80.11 ± 7.62 bpm). In contrast, mean arterial pressure (MAP) was significantly lower during the CMD phase compared to both the baseline and placebo phases (*p* < 0.05). MAP after CMD consumption was 100.33 ± 4.27 mmHg, which was significantly reduced from the baseline value of 107.67 ± 10.77 mmHg (*p* < 0.05). Similarly, MAP during the placebo phase (107.33 ± 3.97 mmHg) remained higher than during the CMD phase, reinforcing the unique impact of CMD in lowering arterial pressure (as shown in Figure 3D).

### 3.6. Effects of CMD Consumption on Blood Lipid Profiles

Figure 4 shows the consequences of CMD consumption on blood lipid profiles. After consuming the CMD and the placebo, subjects experienced the following changes in their blood lipid profiles: The levels of TG were lower during the CMD phase (200.78 ± 53.37 mg/dL) compared to the baseline phase (215.00 ± 90.52 mg/dL) and were statistically lower compared to the placebo phase (260.22 ± 99.98 mg/dL) (*p* < 0.05). There were no statistically significant differences in TC levels between the baseline phase (220.67 ± 44.87 mg/dL) and the CMD phase (200.44 ± 31.30 mg/dL) or the placebo phase (208.67 ± 40.55 mg/dL). The levels of LDL-C did not change significantly during the CMD phase (132.22 ± 21.98 mg/dL) or the placebo phase (134.00 ± 32.88 mg/dL) compared to the baseline period (145.33 ± 36.35 mg/dL). However, both the CMD and placebo phases showed a trend of reducing LDL-C levels. HDL-C levels were similar across all three phases, with no statistically significant differences. The HDL-C levels were recorded as 42.89 ± 8.84 mg/dL during the baseline, 42.22 ± 9.81 mg/dL during the CMD phase, and 42.78 ± 11.62 mg/dL during the placebo phase.

### 3.7. Effects of CMD Consumption on Glucose Homeostasis

The effects of CMD consumption on FPG, FPI, and insulin resistance as measured by the HOMA-IR are illustrated in Figure 5. During the baseline phase, participants exhibited FPG levels of 108.89 ± 16.62 mg/dL. These levels were lower than those recorded during the CMD phase (112.22 ± 33.74 mg/dL) and significantly lower than the levels during the placebo phase (125.72 ± 42.11 mg/dL) (*p* < 0.05). Additionally, subjects’ FPG levels during the placebo phase were significantly higher than during the CMD phase (*p* < 0.05). Moreover, FPI levels were significantly elevated during both the CMD and placebo phases compared to baseline (*p* < 0.05). During the CMD phase, FPI levels were 18.12 ± 8.79 IU/mL, and during the placebo phase, levels were 17.62 ± 4.69 IU/mL. Both values were significantly higher than the baseline FPI levels of 11.38 ± 6.29 IU/mL (*p* < 0.05).

HOMA-IR, a marker of insulin resistance, showed significant changes across the study phases. During the placebo phase, HOMA-IR increased to 5.58 ± 2.50, which was significantly higher than during both the CMD phase (5.03 ± 2.80) and the baseline phase (2.97 ± 1.39) (*p* < 0.05). Although HOMA-IR values during the CMD phase were higher than baseline, the increase was not statistically significant.

### 3.8. Effects of CMD Consumption on Inflammation Marker and Blood Coagulation Indicators 

Figure 6A illustrates the effects of CMD consumption on inflammation markers as measured by serum C-reactive protein (CRP) levels. Serum CRP levels were decreased to 4.04 ± 3.00 mg/L during the CMD phase, a level that was lower than the baseline level of 5.07 ± 5.34 mg/L but not statistically significant. Importantly, CRP levels during CMD consumption were significantly lower than during the placebo phase, where levels increased to 5.98 ± 3.56 mg/L (*p* < 0.05).

No statistically significant differences were observed in blood coagulation indicators among the three phases: baseline, CMD, and placebo. Figure 6B–D illustrate prothrombin time (PT), activated partial thromboplastin time (APTT), and the international normalized ratio (INR), respectively. PT values recorded were 11.03 ± 0.47 s at baseline, 11.61 ± 1.36 s during the CMD phase, and 11.26 ± 1.28 s during the placebo phase. At baseline, APTT values were 27.04 ± 2.45 s, during CMD consumption they were 27.73 ± 2.86 s, and during the placebo phase, they were 28.13 ± 3.26 s. INR values exhibited stability throughout all phases, recorded at 1.00 ± 0.04 during baseline, 1.06 ± 0.13 during the CMD phase, and 1.02 ± 0.12 during the placebo phase.

## 4. Discussion

Mulberries (*Morus* spp.) include three primary species, red (*Morus rubra*), black (*Morus nigra*), and white (*Morus alba*), each with distinct phytochemical profiles and health benefits. Red mulberries, rich in anthocyanins, exhibit strong antioxidant and anti-inflammatory properties [36]. Black mulberries contain high levels of resveratrol, known for its neuroprotective and anti-cancer effects [36,37]. White mulberries, although lower in anthocyanins, are abundant in flavonoids and alkaloids that contribute to hypoglycemic and lipid-lowering activities [36,38]. The KPS-MB-42-1 white mulberry variety was selected for this study due to its high yield, adaptability, and favorable phytochemical profile, including significant phenolic and flavonoid content. These bioactive compounds are associated with antioxidant and anti-inflammatory effects, making this cultivar suitable for developing functional health beverages [39].

The CMD derived from KPS-MB-42-1 exhibits a total phenolic content of 1041.90 mg per 100 g and anthocyanin content of 35.34 mg per 100 g. These phenolic compounds, with potent antioxidant properties, help neutralize free radicals and reduce the risk of chronic diseases, including cardiovascular disorders and certain cancers [20,21]. While white mulberries generally contain lower anthocyanin levels than red or black mulberries [36], CMD’s anthocyanin levels enhance its functional properties, offering potential benefits such as anti-inflammatory effects and improved cognitive health [40]. These findings align with studies reporting phenolic contents in white mulberries ranging from 181 to 1422 mg GAE per 100 g, depending on cultivar and conditions [41]. Additionally, total anthocyanin levels in mulberry varieties have been reported to range from 28 to 185 mg per 100 g fresh weight, with black mulberries exhibiting the highest concentrations. Specifically, anthocyanins such as cyanidin 3-O-rutinoside (60%) and cyanidin 3-O-glucoside (38%) are predominant in mulberry pigments [42]. CMD’s anthocyanin content, while consistent with these studies, underscores the effectiveness of the concentration process in preserving these beneficial compounds, further enhancing the drink’s functional and nutraceutical potential.

Moreover, the sugar profile of the CMD includes fructose (2.51 g/100 g), glucose (3.07 g/100 g), and maltose (13.16 g/100 g). However, it is important to note that the detection of maltose could result from the addition of maltitol, a sugar alcohol used to enhance sweetness and reduce the overall glycemic load of the product [43]. Maltitol, commonly used in health-oriented beverages, is partially hydrolyzed during processing, potentially leading to the presence of maltose as an intermediate product [43,44]. This formulation aligns with trends favoring reduced sugar intake while maintaining sweetness. The addition of maltitol and low natural sugar levels make CMD suitable for individuals with dietary restrictions, such as diabetes or lactose intolerance [43,44]. Its negligible sucrose content supports its role as a low-glycemic alternative, while retaining a superior bioactive profile, positioning CMD as an ideal functional beverage for health-conscious consumers.

The study of the body composition data indicates that CMD consumption did not lead to significant changes in parameters such as body weight, BMI, muscle mass, fat mass, fat percentage, visceral fat area, WC, HC, and WHR when compared to both baseline and placebo consumption. This stability can be attributed to the consistent daily energy and macronutrient intake, as well as the similar energy distribution from carbohydrates, fats, and proteins across all intervention periods. Additionally, the predominantly sedentary lifestyle of the participants, with around 90% reporting no regular exercise or sports activity, likely limited the potential for any notable changes in body composition. Research on mulberry fruit extracts highlights their potential to influence lipid metabolism and reduce fat accumulation, particularly in controlled environments [45]. For instance, animal studies have shown that mulberry fruit extracts, rich in anthocyanins and other bioactive compounds, can improve lipid profiles and reduce fat accumulation in high-fat diet-induced obese mice [46]. These effects are likely driven by the modulation of lipid metabolism pathways and enhanced fat oxidation. Human studies, however, provide mixed evidence. Clinical trials have suggested that mulberry fruit consumption may benefit weight management primarily through glucose metabolism regulation and appetite suppression rather than direct reductions in fat mass [47,48]. For example, Sirikanchanarod et al. demonstrated that mulberry fruit consumption significantly reduced total cholesterol and low-density lipoprotein (LDL) cholesterol levels in hypercholesterolemic individuals but had minimal impact on body weight [48]. These modest outcomes indicate that dietary or physical activity modifications may be necessary to amplify the weight-related benefits of mulberry fruit consumption.

The consumption of CMD significantly reduced SBP, DBP, and MAP compared to baseline levels. Additionally, these parameters remained significantly lower than those observed during the placebo phase. The antihypertensive effects of mulberry products have been documented in previous studies [49,50]. For instance, Hao et al. [50] demonstrated that mulberry anthocyanins improved endothelial function and reduced blood pressure in hypertensive rat models, mediated by increased nitric oxide (NO) bioavailability and reduced oxidative stress [50,51]. In this study, the significant reduction in MAP, which reflects overall blood flow resistance, underscores CMD’s potential to improve vascular function. The bioactive compounds in CMD, particularly anthocyanins, likely contribute to the modulation of vascular tone and reduction in oxidative stress, key factors in hypertension management [52,53]. These results suggest that CMD could be a promising functional beverage for supporting cardiovascular health, especially in individuals at risk of hypertension.

Despite the lack of statistically significant changes in TC, LDL-C, and HDL-C, the significant reduction in TG levels during the CMD phase compared to the placebo phase is notable. However, TG levels during the CMD phase did not significantly differ from baseline. This suggests that CMD consumption may help mitigate TG elevation, especially when compared to the placebo phase, but its direct impact on baseline TG levels remains limited. The observed reduction in TG levels aligns with previous findings on the lipid-lowering effects of mulberry products [54]. Chaiwong et al. demonstrated that dried mulberry powder reduced TG levels and increased HDL-C levels in animal models [55]. Similarly, Peng et al. (2011) reported that mulberry water extracts at 1% and 2% concentrations inhibited TG increases in high-fat-diet-fed rats [56]. Furthermore, Yan et al. (2016) found that supplementing high-fat diets with 5% or 10% freeze-dried mulberry powder significantly reduced TG levels while increasing HDL-C levels [57]. These studies suggest that mulberry bioactive compounds, such as polysaccharides, anthocyanins, and flavonoids, play a role in regulating TG metabolism, particularly under conditions of dietary stress like high-fat diets [54,55,56,57].

Insulin resistance is characterized by reduced cellular responsiveness to insulin, leading to elevated insulin secretion and fasting blood glucose levels [57,58]. Studies have shown that higher BMI is associated with increased FPI and inflammation, which exacerbate insulin resistance [59]. In this study, the significant reduction in FPG levels during the CMD phase compared to the placebo phase suggests that CMD may have helped stabilize glucose metabolism. The findings align with prior research on mulberry’s potential antidiabetic effects [57,58]. Wattanathorn et al. demonstrated that mulberry fruit extracts at doses of 50 mg/kg and 250 mg/kg improved fasting blood glucose levels in animal models consuming high-carbohydrate and high-fat diets [58]. Similarly, Yan et al. reported that anthocyanin extracts from mulberry effectively inhibited glucose absorption in HepG2 cells and reduced glucose levels in vivo. These studies highlight the role of mulberry compounds in improving glucose homeostasis by regulating carbohydrate metabolism and insulin sensitivity [57]. The significant increase in FPI levels during the CMD phase suggests enhanced insulin secretion, likely as a compensatory response to maintain glucose levels. While HOMA-IR values were higher during the CMD phase compared to baseline, they remained significantly lower than during the placebo phase. This finding indicates that CMD consumption may partially alleviate insulin resistance, possibly through its anti-inflammatory and antioxidant properties, which reduce oxidative stress and improve insulin signaling pathways [57,60].

CRP is a liver-synthesized protein that increases during inflammation and is elevated in individuals with obesity due to higher adipose tissue levels [61]. Elevated CRP impairs leptin function, potentially disrupting satiety signals and promoting metabolic dysregulation, including insulin resistance [61,62]. Participants in this study exhibited baseline CRP levels of 5.07 ± 5.34 mg/L, indicative of low-grade inflammation, which is commonly associated with obesity [59]. The reduction in CRP during CMD consumption, relative to placebo, suggests that CMD’s high total phenolic compounds (1041.9 mg/100 g) may have mitigated inflammatory responses. As reported by Rodriguez-Ramiro et al. [63] and Simioni et al. [27], phenolic compounds are potent antioxidants that reduce oxidative stress and inflammation. CMD’s polyphenolic composition likely neutralized free radicals, reducing inflammatory cytokines and subsequently lowering CRP levels [27,63]. The significant reduction in CRP during the CMD phase compared to placebo underscores CMD’s role in alleviating inflammation, making it a promising intervention for managing chronic inflammation in obesity.

While CRP reductions suggest anti-inflammatory effects, no significant changes were observed in coagulation markers such as PT, APTT, and INR across all phases. The coagulation parameters remained within normal ranges (PT: 11.03–11.61 s; APTT: 27.04–28.13 s; INR: 1.00–1.06) [64]. These findings indicate that CMD consumption does not disrupt hemostatic balance, which is crucial in populations at risk of thrombotic complications, such as individuals with metabolic syndrome or obesity [65,66]. The clinical literature highlights the association between obesity, inflammation, and hypercoagulability, with shortened APTT and elevated CRP levels linked to increased thrombotic risk [67,68]. In this study, normal coagulation times and unchanged PT and APTT values suggest that CMD consumption neither exacerbated nor alleviated hypercoagulability. This finding is consistent with studies indicating that polyphenol-rich foods may primarily exert their effects on inflammation rather than directly influencing coagulation parameters [69].

This study was designed as a randomized, single-blind clinical trial. Fifteen participants were randomized to receive either CMD or a placebo in each allocation period. Each allocation period lasted 6 weeks, with an additional 2-week washout period, resulting in a total study duration of approximately 14 weeks for each participant. The long study duration posed challenges in maintaining participant adherence, leading to a dropout rate of approximately 20%. Ultimately, only 12 participants completed this study and provided analyzable data. This dropout rate exceeded initial expectations, highlighting a limitation in the planning phase, as the anticipated dropout percentage was underestimated. This limitation underscores the importance of implementing robust participant retention strategies and accounting for potential dropout rates in future studies, particularly when long-term participation is required. Despite this limitation, the findings of this study provide valuable preliminary data that can serve as a foundation for future investigations with larger sample sizes and optimized study designs to enhance the reliability and generalizability of the results.

CMD demonstrated potential as a health-promoting beverage, showing benefits for cardiovascular and metabolic health without adverse effects. Its anti-inflammatory properties, likely attributed to its high polyphenol content, and the absence of side effects over the 6-week intervention period underscores its safety and tolerability. Future studies should focus on larger, more diverse populations, extended intervention durations, and varying dosages to confirm these findings and further explore CMD’s long-term effects on metabolic and cardiovascular risk factors. Additionally, investigating CMD’s mechanisms of action and potential synergistic effects with dietary or lifestyle interventions could enhance its efficacy and applicability as a functional health beverage.

## 5. Conclusions

This study demonstrates that CMD derived from the Kamphaeng Saen mulberry (KPS-MB-42-1) cultivar significantly improves metabolic and cardiovascular markers in individuals with obesity. CMD consumption notably reduced systolic and diastolic blood pressure, mean arterial pressure, triglycerides, and fasting plasma glucose levels while lowering C-reactive protein levels, indicating its potential anti-inflammatory effects. These findings support CMD as a promising health drink for managing metabolic syndrome and associated chronic diseases. Further studies with larger sample sizes are recommended to confirm these benefits and explore long-term effects.

## Figures and Tables

**Figure 1 nutrients-16-04336-f001:**
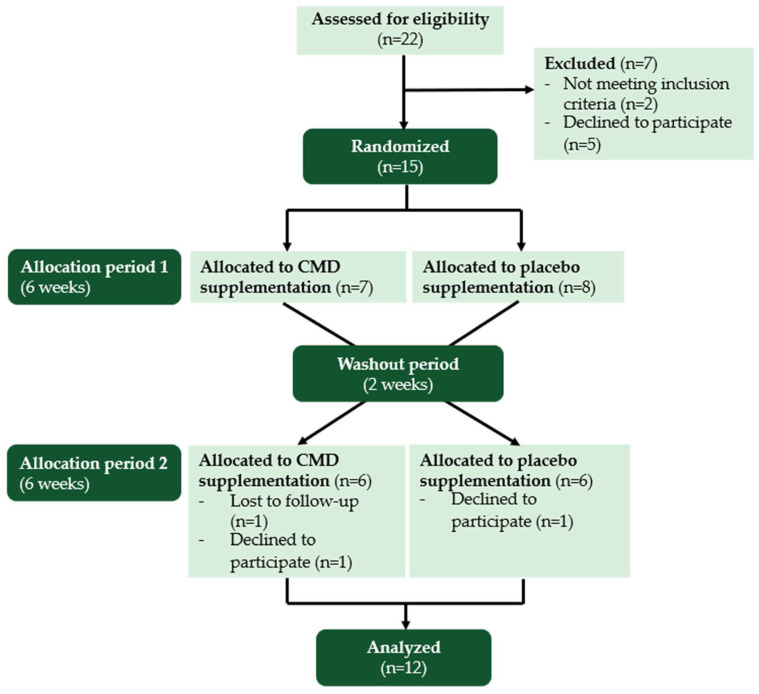
Flowchart of subjects in the randomized, single-blind, and crossover trial.

**Figure 2 nutrients-16-04336-f002:**
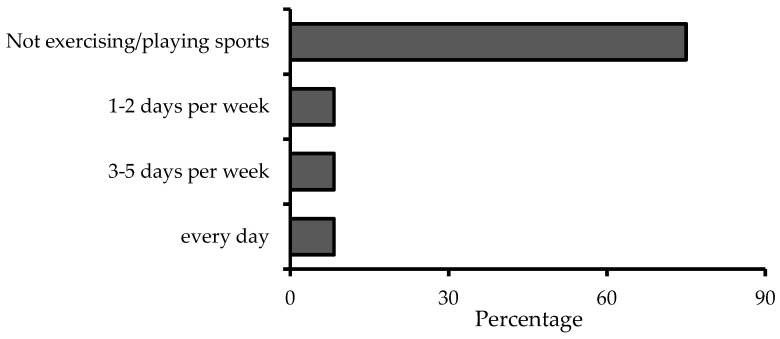
Frequency of exercise or sports of subjects (n = 12).

**Figure 3 nutrients-16-04336-f003:**
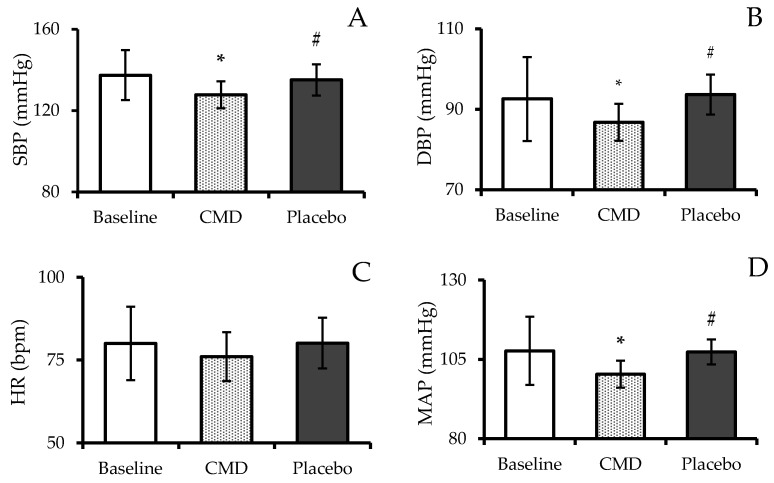
Systolic blood pressure, SBP (**A**); diastolic blood pressure, DBP (**B**); heart rate, HR (**C**); and mean arterial pressure, MAP (**D**) of subjects after receiving intervention at each period. The results are reported as mean ± SD (n = 12). * shows a statistically significant difference when compared with baseline (*p*-value < 0.05), and # shows a statistically significant difference when compared with CMD (*p*-value < 0.05).

**Figure 4 nutrients-16-04336-f004:**
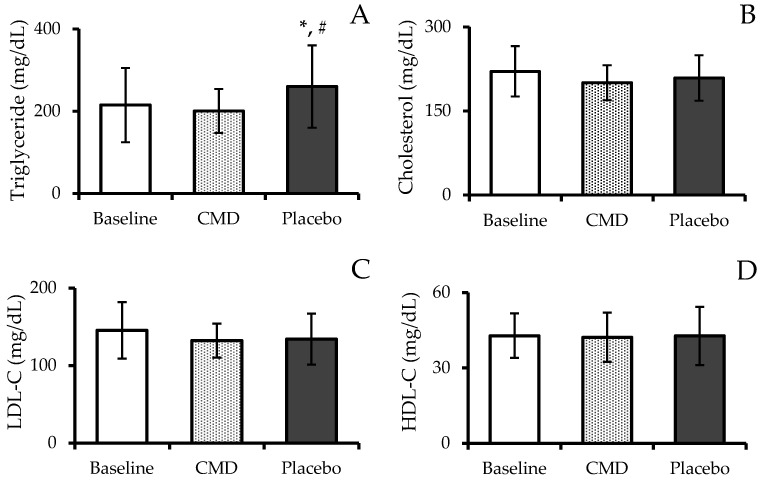
Triglyceride (**A**); cholesterol (**B**); low-density lipoprotein cholesterol, LDL-C (**C**); and high-density lipoprotein cholesterol, HDL-C (**D**) of subjects after receiving intervention at each period. The results are reported as mean ± SD (n = 12). * shows a statistically significant difference when compared with baseline (*p*-value < 0.05), and # shows a statistically significant difference when compared with CMD (*p*-value < 0.05).

**Figure 5 nutrients-16-04336-f005:**
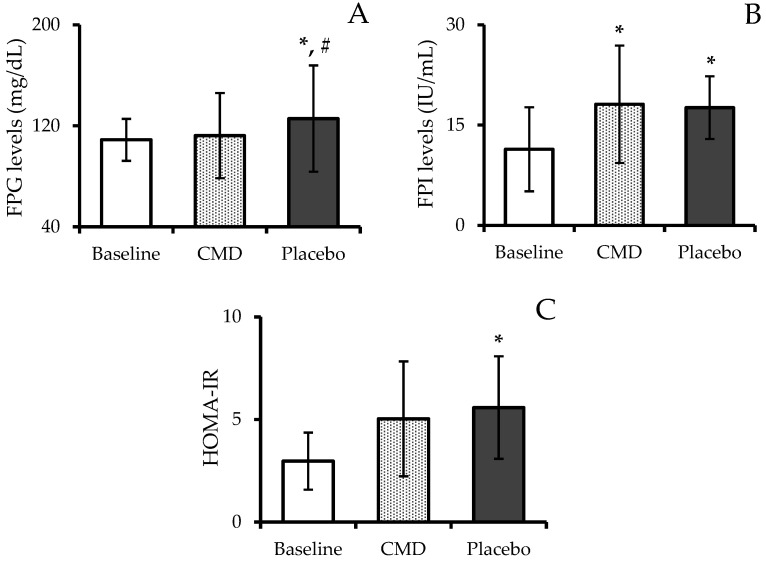
Fasting plasma glucose (FPG) levels (**A**); fasting plasma insulin (FPI) levels (**B**); and homeostatic model assessment of insulin resistance, HOMA-IR (**C**) of subjects after receiving intervention at each period. The results are reported as mean ± SD (n = 12). * shows a statistically significant difference when compared with baseline (*p*-value < 0.05), and # shows a statistically significant difference when compared with CMD (*p*-value < 0.05).

**Figure 6 nutrients-16-04336-f006:**
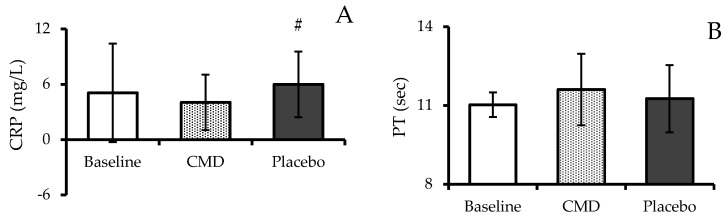
C-reactive protein (CRP) levels (**A**); prothrombin time, PT (**B**); activated partial thromboplastin time, APTT (**C**); and the international normalized ratio, INR (**D**) of subjects after receiving intervention at each period. The results are reported as mean ± SD (n = 12). # shows a statistically significant difference when compared with CMD (*p*-value < 0.05).

**Table 1 nutrients-16-04336-t001:** Nutritional values, sugar profile, and bioactive compounds of CMD.

Items	Amount per 100 g
**Nutritional values**	
Ash (g)	0.45
Fiber (g)	<0.10
Calories (Kcal)	78.08
Calories from fat (Kcal)	1.08
Carbohydrate (g)	18.78
Fat (g)	0.12
Moisture (g)	80.18
Protein (%N × 6.25) (g)	0.47
Starch (g)	0.0035
Vitamin C (mg)	<0.015
**Sugar profiles**	
Fructose (g)	2.51
Glucose (g)	3.07
Sucrose (g)	Not Detected
Maltose (g)	13.16
Lactose (g)	Not Detected
Total sugar (g)	18.74
**Bioactive compounds**	
Total phenolic compounds (mg)	1041.90
Total anthocyanins (mg)	35.34

**Table 2 nutrients-16-04336-t002:** Baseline physical characteristics of subjects.

Physical Characteristics	Mean ± SD (n = 12)	Range
Age (years)	46.57 ± 8.49	30–55
Body weight (kg)	92.1 ± 15.55	74.2–123.2
Body mass index, BMI (kg/m^2^)	32.1 ± 5.98	25.1–38.0
Muscle mass (kg)	33.7 ± 6.45	25.5–43.0
Fat mass (kg)	32.2 ± 12.26	13.3–50.5
Fat (%)	34.4.2 ± 10.07	17.2–50.1
Visceral fat area (level)	13.89 ± 5.06	5–20
Waist circumference, WC (cm)		
Male	110.83 ± 16.80	90–141
Female	110.00 ± 30.33	91–127
Hip circumference, HC (cm)		
Male	112.25 ± 15.99	97–143
Female	119.50 ± 31.17	106–129
Waist-to-hip ratio, WHR		
Male	0.99 ± 0.03	0.93–1.02
Female	0.98 ± 0.05	0.86–0.98
Heart rate (beats per min, bpm)	80 ± 11.12	55–92
Systolic blood pressure, SBP (mmHg)	137.4 ± 12.28	115–151
Diastolic blood pressure, DBP (mmHg)	92.56 ± 10.45	73–106

**Table 3 nutrients-16-04336-t003:** Baseline blood chemistry of subjects.

Blood Chemistry Tests	Mean ± SD (n = 12)	Range	Reference Range
Triglyceride (mg/dL)	215.00 ± 90.52	66–340	<150
Cholesterol (mg/dL)	220.67 ± 44.87	176–310	<200
LDL-C (mg/dL)	145.33 ± 36.35	99–202	<130
HDL-C (mg/dL)	42.89 ± 8.84	33–58	>40
Fasting plasma glucose, FPG (mg/dL)	108.89 ± 16.62	92–142	70–110
Fasting plasma insulin, FPI (IU/mL)	11.38 ± 6.29	2.6–22.8	3–25
HOMA-IR	2.97 ± 1.39	0.7–5.2	0.5–1.4
Blood urea nitrogen, BUN (mg/dL)	13.38 ± 4.17	6–19	Male 8–26
			Female 7–20
Creatinine (mg/dL)	0.93 ± 0.30	0.62–1.50	Male 0.73–1.18
			Female 0.55–1.02

**Table 4 nutrients-16-04336-t004:** The amount of nutrients and energy that subjects received during the intervention period.

Nutrients and Energy	Daily Nutrient Intake (Mean ± SD, n = 12)		
	Baseline	CMD ^nd1^	Placebo ^nd1, nd2^
Energy (Kcal)	1647.73 ± 335.27	1721.19 ± 302.26	1669.83 ± 358.18
Carbohydrate (g)	236.28 ± 76.20	236.71 ± 68.87	232.35 ± 73.04
Fat (g)	49.38 ±7.61	55.95 ± 11.34	54.23 ± 13.07
Protein (g)	64.55 ± 12.29	67.70 ± 14.14	63.08 ± 12.18
Vitamin A (µg RAE)	214.93 ± 150.01	439.88 ± 482.26	307.70 ± 377.70
Vitamin B1 (mg)	1.27 ± 0.97	1.64 ± 0.52	1.56 ± 0.71
Vitamin B2 (mg)	0.87 ± 0.32	1.37 ± 1.30	1.27 ± 1.11
Vitamin C (mg)	26.71 ± 14.51	37.04 ± 26.53	24.82 ± 18.40
Niacin (mg)	14.05 ± 4.90	16.99 ± 7.53	16.35 ± 5.30
Calcium (mg)	384.15 ± 182.25	363.25 ± 280.40	426.50 ± 36.034
Iron (mg)	8.84 ± 3.52	9.1 ± 4.22	8.58 ± 3.03

^nd1^ There was no significant difference between baseline and post-CMD/placebo consumption. ^nd2^ There was no significant difference between post-CMD and post-placebo consumption.

**Table 5 nutrients-16-04336-t005:** Distribution of energy-providing nutrients that subjects received at each time point during the intervention.

Macronutrients	Percentage Distribution of Energy-Providing Nutrients (Mean ± SD, n = 12)		
	Baseline	CMD ^nd1^	Placebo ^nd1, nd2^
Carbohydrate	56.91 ± 7.94	54.60 ± 7.35	55.13 ± 8.61
Fat	27.14 ± 5.25	29.30 ± 6.33	29.51 ± 7.99
Protein	15.94 ± 3.57	16.11 ± 3.49	15.37 ± 2.08

^nd1^ There was no significant difference between baseline and post-CMD/placebo consumption. ^nd2^ There was no significant difference between post-CMD and post-placebo consumption.

**Table 6 nutrients-16-04336-t006:** The body composition of the subjects at different intervention periods.

Variables	Body Composition (Mean ± SD, n = 12)		
	Baseline	CMD ^nd1^	Placebo ^nd1, nd2^
Body weight (kg)	94.86 ± 16.54	95.54 ± 16.88	95.94 ± 15.87
Body mass index, BMI (kg/m^2^)	33.51 ± 6.06	33.74 ± 6.12	33.87 ± 5.79
Muscle mass (kg)	33.07 ± 7.31	32.84 ± 7.60	33.37 ± 7.59
Fat mass (kg)	37.64 ± 8.25	37.94 ± 9.06	38.59 ± 9.18
Fat (%)	35.93 ± 10.78	36.57 ± 11.05	37.10 ± 11.94
Visceral fat area (level)	15.57 ± 4.04	15.57 ± 4.31	15.71 ± 4.39
Waist circumference, WC (cm)			
Male	110.83 ± 16.80	108.50 ± 15.63	110.50 ± 17.37
Female	110.00 ± 18.08	111.00 ± 15.00	117.00 ± 10.53
Hip circumference, HC (cm)			
Male	112.33 ± 16.18	112.00 ± 15.17	113.33 ± 15.27
Female	119.67 ± 12.09	119.00 ± 10.44	120.33 ± 10.78
Waist to hip ratio, WHR			
Male	0.99 ± 0.03	0.96 ± 0.04	0.97 ± 0.04
Female	0.91 ± 0.06	0.93 ± 0.05	0.97 ± 0.04

^nd1^ There was no significant difference between baseline and post-CMD/placebo consumption. ^nd2^ There was no significant difference between post-CMD and post-placebo consumption.

## Data Availability

The data presented in this study are available upon request from the corresponding author. The data are not publicly available due to the raw data might contain identifiable personal information of the participants. Sharing such data would require approval from the ethics committee and explicit consent from the participants.

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
