# Peer review of "Nutraceutical Properties of Thai Mulberry (*Morus alba* L.) and Their Effects on Metabolic and Cardiovascular Risk Factors in Individuals with Obesity: A Randomized, Single-Blind Crossover Trial"

_nutrients, 2024, doi:10.3390/nu16244336_

Round 1

Reviewer 1 Report

Comments and Suggestions for Authors

The article "Nutraceutical properties of Thai mulberry (Morus alba L.) on metabolic and cardiovascular risk factors in individuals with obesity: a randomized, single-blind, crossover trial" contains original results on the effects of concentrated mulberry drink (CMD) on body composition, blood pressure, and blood sample parameters. However, for a complete understanding and confirmation of the role of mulberries in these parameters, it is necessary to clarify certain aspects:

Introduction: The introduction is briefly and clearly described, providing a good introduction to the research topic.

Materials and Methods: Have you measured any parameters, such as oxidative stress (antioxidant capacity, polyphenols, etc.), to confirm that changes have actually occurred as a result of CMD consumption?

Statistical analysis: Considering that this is a human study, have you determined the test's power and the minimum number of participants that must be included in the research to ensure the significance of the results and the relevance of the study?

Results: Figure 6. What do you consider to be the cause of the large standard deviation in the baseline group? 

Disscusion: Add the limitations of the study.

Author Response

Reply: The authors sincerely thank the reviewers for their insightful questions and valuable suggestions, which have greatly contributed to improving our manuscript. These suggestions have not only enhanced the quality of this manuscript but also provided valuable insights that will guide the development of our future studies. All questions have been addressed, as detailed below.

Introduction: The introduction is briefly and clearly described, providing a good introduction to the research topic.

Reply: Thank you very much for your positive feedback regarding the introduction. We are pleased that you found it clear and appropriately set the stage for the research topic. Your comments are greatly appreciated.

Materials and Methods: Have you measured any parameters, such as oxidative stress (antioxidant capacity, polyphenols, etc.), to confirm that changes have actually occurred as a result of CMD consumption?

Reply: Thank you for your insightful question. Unfortunately, we did not measure oxidative stress markers directly in the participants during this study. However, previous studies have extensively reported the presence of bioactive compounds in mulberries, including total phenolic compounds and total anthocyanins, which play a crucial role in mitigating oxidative stress and are strongly associated with the prevention of chronic diseases (Mattioli et al., 2020; Zhang et al., 2015).

In this study, we monitored the stability of these bioactive compounds in the CMD product after thermal processing. Specifically, CMD retained total phenolic compounds and total anthocyanins at concentrations of 1,041.9 mg/100 g and 35.34 mg/100 g, respectively. This suggests that the bioactive compounds, known for their antioxidative properties, remained present in appreciable quantities, even after processing. Additionally, to indirectly assess the physiological impact, we measured inflammatory markers such as serum C-reactive protein (CRP) levels. CRP is a well-established marker positively correlated with oxidative stress and systemic inflammation (Abramson et al., 2005; Cottone et al., 2006). Changes in CRP levels observed in our study provide valuable insights into the anti-inflammatory potential of CMD, which aligns with the hypothesized antioxidative effects of its bioactive components.

Future studies are warranted to include direct measurements of oxidative stress markers, such as total antioxidant capacity or specific oxidative stress biomarkers, to further confirm the antioxidative impact of CMD consumption in vivo. We appreciate your valuable feedback and look forward to further exploring this aspect in subsequent research.

REFERENCES

Abramson, J.L.; Hooper, W.C.; Jones, D.P.; et al. Association between novel oxidative stress markers and C-reactive protein among adults without clinical coronary heart disease. Atherosclerosis 2005, 178, 115-21. Doi: 10.1016/j.atherosclerosis.2004.08.007.

Cottone, S.; Mulè, G.; Nardi, E.; et al. Relation of C-reactive protein to oxidative stress and to endothelial activation in essential hypertension. Am J Hypertens. 2006, 19, 313-8. Doi: 10.1016/j.amjhyper.2005.09.005.

Mattioli, R.; Francioso, A.; Mosca, L.; et al. Anthocyanins: A Comprehensive Review of Their Chemical Properties and Health Effects on Cardiovascular and Neurodegenerative Diseases. Molecules 2020, 25, 3809. Doi: 10.3390/molecules25173809.

Zhang, Y.-J.; Gan, R.-Y.; Li, S.; et al.  Antioxidant Phytochemicals for the Prevention and Treatment of Chronic Diseases. Molecules 2015, 20, 21138-56. Doi: 10.3390/molecules201219753.

Statistical analysis: Considering that this is a human study, have you determined the test's power and the minimum number of participants that must be included in the research to ensure the significance of the results and the relevance of the study?

Reply: Thank you for raising this important point. The sample size for our study was calculated using the G*Power software (version 3.1.9.4). The calculation was based on detecting an effect size of 0.5, derived from changes in LDL-cholesterol reported in a previous study (Chang et al., 2014). We used a significance level (α) of 0.05 and a desired power of 90%, which yielded a minimum required sample size of 14 participants per group.

For this study, we initially randomized 15 participants to receive either CMD or placebo in each allocation period. However, by the end of the study, only 12 participants had completed the trial and provided analyzable data, resulting in an observed dropout rate of approximately 20%. While the initial sample size calculation ensured adequate power for detecting significant changes, the higher-than-anticipated dropout rate necessitates that the results of this study be interpreted as preliminary. Consequently, this study should be considered a pilot investigation. The findings provide valuable baseline data and insights that will inform the design of future studies with larger sample sizes and more robust statistical power to ensure greater generalizability and relevance.

REFERENCES

Chang, H.C.; Peng, C.H., Yeh, D.M.; et al. Hibiscus sabdariffa extract inhibits obesity and fat accumulation, and improves liver steatosis in humans. Food Funct. 2014, 5, 734-739. Doi: 10.1039/c3fo60495k.

Results: Figure 6. What do you consider to be the cause of the large standard deviation in the baseline group?

Reply: Thank you for your insightful question. The blood chemistry and biochemical analyses in our study were conducted at the Chandrubeksa Hospital Laboratory, Kamphaeng Saen District, Nakhon Pathom Province, which is a certified facility adhering to standard operating procedures. Therefore, the likelihood of analytical errors contributing to the large standard deviation (SD) in the baseline group is minimal. One plausible explanation for the observed large SD is the relatively small sample size in the study. Limited sample sizes can amplify variability, as outliers or individual differences among participants have a more pronounced effect on the statistical measures, leading to higher SD values. This phenomenon is common in pilot or small-scale studies and highlights the need for larger sample sizes in future studies to reduce variability and enhance the robustness of the results.

To improve the clarity of the results, we have updated Figure 6A to include the SD values for serum CRP levels, providing a more comprehensive visualization of the data variability.

Disscusion: Add the limitations of the study.

Reply: Thank you for your valuable suggestion. The limitations of our study are as follows:

This study was conducted as a randomized, single-blind clinical trial. Based on sample size calculations using G*Power software, a minimum of 14 participants was required to achieve adequate statistical power. To account for potential dropouts, we randomized 15 participants to receive either CMD or placebo in each allocation period. Each allocation period lasted 6 weeks, with an additional 2-week washout period between allocations. Consequently, participants were required to remain in the study for approximately 14 weeks. Due to the long period of study duration and potential challenges in maintaining participant adherence over this period, we observed a dropout rate of approximately 20%. Ultimately, only 12 participants completed the study and provided data for final analysis. This dropout rate exceeded our initial expectations and highlights a limitation in our planning, as the anticipated dropout percentage was underestimated.

This limitation underscores the need for careful consideration of participant retention strategies and more robust dropout estimations in future studies, especially when long study durations are required. Despite these limitations, the data from this study provide valuable preliminary insights that can serve as a foundation for future investigations with larger sample sizes and improved study designs.

The authors have incorporated a detailed discussion of these limitations into the manuscript under the Discussion section, in lines 554–567.

Reviewer 2 Report

Comments and Suggestions for Authors

The manuscript presented for the concerns an important problem, which the search for natural dietary supplements supporting the prevention and treatment of civilization-related diseases. Although the subject matter in general is not new, any research that can identify further products/substances of health-promoting importance is vital.

My comments are as follows:

Is the supplement used in the study intended for the commercial use or was it elaborated just for the need of the study?

Methods:

point 2.2. How were the participants recruited for the study? Through advertisements or other means?

lines 147 -149 - How can you relate the study on resveratrol supplementation (reference 32) to present study? How did you conclude that the amount is suitable?

lines 152-153 - Is the study (ref.33) suficient base for establishing supplementation period in the present study?

Figure 1 - scheme of the study - Were physical examinations and blood analyses also conducted after allocation period 1?

Author Response

Reply: The authors sincerely thank the reviewers for their insightful questions and valuable suggestions, which have greatly contributed to improving our manuscript. These suggestions have not only enhanced the quality of this manuscript but also provided valuable insights that will guide the development of our future studies. All questions have been addressed, as detailed below.

Is the supplement used in the study intended for the commercial use or was it elaborated just for the need of the study? 

Reply: Mulberry is a widely cultivated plant in Thailand, primarily grown for fresh fruit consumption or for utilizing the leaves as feed for silkworms. However, commercial-scale cultivation of mulberry fruit remains limited. Typically, the general population consumes mulberries fresh, based on the available yield, as nutritional guidelines generally advocate for the consumption of fresh, minimally processed fruits.

Our study focused on processing mulberries to enhance their nutritional and functional properties. We found that thermal treatment of mulberry fruit (at 70–90°C) resulted in mulberry juice with higher levels of total phenolic compounds compared to non-thermally processed juice. These phenolic compounds are well-documented for their potent antioxidant activities and potential health benefits. Additionally, fresh mulberries are highly perishable due to their high water content, leading to rapid spoilage and limited shelf life. This often results in significant post-harvest waste. To address these challenges, our research aimed to explore the potential of mulberry fruit processing to extend shelf life and enhance usability. The specific product used in this study was developed by the researchers for the purposes of this investigation.

Moreover, to our knowledge, there is currently no existing research examining the metabolic effects of consuming Kamphaeng Saen mulberry (KPS-MB-42-1) in humans, particularly among individuals with obesity. This study provides foundational insights into the potential metabolic benefits of mulberry consumption and highlights the feasibility of developing mulberry-based products for broader nutritional and commercial applications.

Methods: point 2.2. How were the participants recruited for the study? Through advertisements or other means?

Reply: Participants for this study were recruited after receiving approval from the Institutional Review Board (IRB). Recruitment was conducted through the use of posters, which were displayed to announce and invite volunteers interested in participating in the project. All interested individuals were required to meet the inclusion criteria specified by the study to ensure eligibility.

We have incorporated this information into the manuscript in lines 113–116 to provide clarity on the recruitment process.

Methods: lines 147 -149 - How can you relate the study on resveratrol supplementation (reference 32) to present study? How did you conclude that the amount is suitable?

Reply: We understand the concern raised in your question and acknowledge the referencing error on our part. The cited study on resveratrol supplementation (reference 32) was not appropriately aligned with the current study. Several studies have demonstrated health benefits associated with varying amounts of total phenolic compounds. To ensure consistency with the present study, we have revised the reference to the study by Cassidy et al. (2013), which found that 35 mg of total anthocyanins in berries aligns with the levels used in this research and has been shown to improve metabolic markers.

This revision has been incorporated into the manuscript, specifically in lines 151–153, to provide a more accurate and relevant context. We appreciate your observation and the opportunity to enhance the clarity and accuracy of our manuscript.

REFERENCES

Cassidy, A.; Mukamal, K.J.; Liu, L.; et al. High anthocyanin intake is associated with a reduced risk of myocardial infarction in young and middle-aged women. Circulation 2013, 127, 188–196. Doi: 10.1161/CIRCULATIONAHA.112.122408.

Methods: lines 152-153 - Is the study (ref.33) suficient base for establishing supplementation period in the present study?

Reply: We understand your concern regarding the sufficiency of the referenced study (ref. 33) as a basis for establishing the supplementation period in the present study. The cited reference pertains to research on pomegranate juice. While pomegranate juice and mulberrie drink share similarities in their antioxidant properties, particularly due to anthocyanins and phenolic compounds, they also exhibit differences in specific bioactive components.

To strengthen the rationale for the supplementation period, we have included additional references that investigate the effects of berry-based interventions. Notably, studies on a blueberry smoothie (Stull et al., 2015) and a strawberry beverage (Amani et al., 2014) demonstrated significant health benefits in metabolic variables over a supplementation period of 6 weeks. These findings provide a more robust foundation for selecting the intervention duration in the current study.

We appreciate your observation and have revised the manuscript accordingly to incorporate these additional references.

REFERENCES

Amani, R.; Moazen, S.; Shahbazian, H.; et al. Flavonoid-rich beverage effects on lipid profile and blood pressure in diabetic patients. World J Diabetes. 2014, 5, 962–968. Doi: 10.4239/wjd.v5.i6.962.

Stull, A.J.; Cash, K.C., Champagne, C.M.; et al. Blueberries improve endothelial function, but not blood pressure, in adults with metabolic syndrome: a randomized, double-blind, placebo-controlled clinical trial. Nutrients 2015, 7, 4107–4123. Doi: 10.3390/nu7064107.

Methods: Figure 1 - scheme of the study - Were physical examinations and blood analyses also conducted after allocation period 1?

Reply: Physical examinations and blood analyses were conducted during each allocation period in the same manner as at baseline. This has been detailed in the manuscript, specifically in lines 157–161, to ensure clarity regarding the study procedures.
